# Association between intraoperative fluid balance, vasopressors and graft complications in liver transplantation: A cohort study

Jordan Larivière[1], Jeanne-Marie Giard[2], Rui Min Zuo[3], Luc Massicotte[4], Michaël Chassé[5], François Martin Carrier[3,5,6]*

1 Department of Medicine, Université de Montréal, Montréal, Québec, Canada, 2 Department of Medicine— Liver Diseases Division, Centre Hospitalier de l'Université de Montréal (CHUM), Montréal, Québec, Canada, 3 Faculty of Medicine, Université de Montréal, Montréal, Québec, Canada, 4 Department of Anesthesiology, Centre Hospitalier de l'Université de Montréal (CHUM), Montréal, Québec, Canada, 5 Department of Medicine–Intensive Care Division, Centre Hospitalier de l'Université de Montréal (CHUM), Montréal, Québec, Canada, 6 Carrefour de l'innovation, Centre de Recherche du Centre Hospitalier de l'Université de Montréal (CRCHUM), Montréal, Québec, Canada

* francois.martin.carrier@umontreal.ca

**Data Availability Statement:** Due to national regulations in the Province of Quebec (Canada) and restrictions from the Centre de recherche du Centre

## Abstract

### Introduction

Biliary complications following liver transplantation are common. The effect of intraoperative fluid balance and vasopressors on these complications is unknown.

### Materials and methods

We conducted a cohort study between July 2008 and December 2017. Our exposure variables were the total intraoperative fluid balance and the use of vasopressors on ICU admission. Our primary outcome was any biliary complication (anastomotic and non-anastomotic strictures) up to one year after transplantation. Our secondary outcomes were vascular complications, primary graft non-function and survival.

### Results

We included 562 consecutive liver transplantations. 192 (34%) transplants had a biliary complication, 167 (30%) had an anastomotic stricture and 56 had a non-anastomotic stricture (10%). We did not observe any effect of intraoperative fluid balance or vasopressor on biliary complications (HR = 0.97; 95% CI, 0.93 to 1.02). A higher intraoperative fluid balance was associated with an increased risk of primary graft non-function (non-linear) and a lower survival (HR = 1.40, 95% CI, 1.14 to 1.71) in multivariable analyses.

### Conclusion

Intraoperative fluid balance and vasopressors upon ICU admission were not associated with biliary complications after liver transplantation but may be associated with other

hospitalier de l'Université de Montréal Review Ethic Board (REB), health medical data cannot be made available publicly. However, complete access to the research dataset is possible for research purpose after appropriate privacy agreements between research parties have been completed. Data access requests may be sent to the corresponding author (francois.martin.carrier@umontreal.ca), or directly to our REB (ethique.recherche.chum@ssss.gouv.qc.ca). Any further information may be provided by our REB head (marie.josee.bernardi.chum@ssss.gouv.qc.ca). The study reported in this manuscript is registered with the study number 18.231 and complied to all local regulations. The dataset is named "TL3_clean.RData" and is localized in our local research directories.

**Funding:** This work was funded by the Centre de recherche du CHUM (all authors). MC and FMC are recipients of a Career Award from the Fonds de la Recherche du Québec – Santé. The funders had no role in study design, data collection and analysis, decision to publish, or preparation of the manuscript.

**Competing interests:** The authors have declared that no competing interests exist.

**Abbreviations:** AS, Anastomotic Stricture; CI, Confidence Intervals; CIT, Cold Ischemia Time; CTA, Computed Tomography Angiography; CVP, Centre Venous Pressure; DCD, Donation after Circulatory Death; DRI, Donor Risk Index; GEE, Generalized Estimating Equations; HAS, Hepatic Artery Stenosis; HAT, Hepatic Artery Thrombosis; HR, Hazard Ratio; ICU, Intensive Care Unit; LT, Liver Transplantation; MELD, Model for End-stage Liver Disease; NAS, Non-Anastomotic Stricture; NDD, Neurological Determination of Death; OR, Odds Ratio; PNF, Primary Graft Non-Function; RBC, Red Blood Cells; RRT, Renal Replacement Therapy.

adverse events. Intraoperative hemodynamic management must be prospectively studied to further assess their impact on liver recipients' outcomes.

## Introduction

Liver transplantation (LT) is the end-of-line treatment for end-stage liver disease [1]. Even though the annual number of LT is increasing, the number of patients on waiting lists remains high [2]. With up to 10 percent of grafts being non-functional at one year [2], interventions to improve their survival and resource utilization are needed.

Biliary graft anastomotic and non-anastomotic strictures are frequent after LT, with reported incidences varying between 10 and 30% [3–14]. They are associated with late graft non-function and a poor quality of life [15]. These complications lead to multiple invasive procedures, resulting in increased costs and a higher morbidity [16]. Decreasing such complications, the Achilles heel of liver transplantation, and increasing graft survival should be the focus of research.

Ischemic injury is believed to be the main etiological mechanism of non-anastomotic strictures (NAS), as suggested by the high incidence following hepatic artery thrombosis [9]. This postulated etiology has led to the name "ischemic cholangiopathy" or more recently, "post-transplant cholangiopathy" [9]. Immune-mediated injury has also been suggested as another potential mechanism, especially in late presentation of the disease [9,17]. Technical factors and recipient-donor bile duct mismatch are recognized to play important roles in anastomotic strictures (AS) development [18]. However, local ischemic and immunologic factors are also reported to contribute to the incidence of AS [16,19]. Similarly, intraoperative fluid management has been associated with digestive tract anastomosis complications, suggesting that both technical and perfusion factors may contribute to surgical anastomosis complications [20,21].

Most factors associated with graft complications seem to be related to recipients' postoperative course [12,22] or donor [23] and graft procurement characteristics (agonal time in donors after cardiocirculatory death (DCD), cold ischemia time and the type of preservation solution) [9,13,17]. Few perioperative variables have been identified to increase such complications. A correlation between patient's cardiac output and graft blood flow has been suggested, although no association has been clearly established with graft complications [24]. Since biliary ducts are sensitive to ischemic injury, a prolonged period of reduced graft flow might increase all biliary graft complications [9]. This suggests that intraoperative hemodynamic management, through effects on recipients' cardiac output and graft perfusion, could be associated with graft complications, a poorly explored association in liver transplantation.

The primary objective of this cohort study was to evaluate the association between intraoperative fluid balance and vasopressor use on all postoperative graft biliary strictures in one LT center. The secondary objectives were to explore the association between the same exposure variables and non-anastomotic biliary strictures, anastomotic biliary strictures, graft hepatic artery complications, primary graft non-function and survival. Our hypothesis was that a lower fluid balance and the need for vasopressor at the end or surgery would be associated with a higher risk of biliary complications.

## Material and methods

### Study design and settings

We report this study according to STROBE guidelines [25]. After study approval by our local research ethics board, we conducted a retrospective observational cohort study at the Centre

Hospitalier de l'Université de Montréal (CHUM), a liver transplantation center performing around 70 transplantations per year. Our research ethics board waived the need for consent for retrospective data collection as per national regulations.

Donors may have been previously registered in one of the national organ donor registries if requested by patients (*Régie de l'assurance maladie du Québec* or the *Chambre des notaires du Québec* organ donor registry (https://www.transplantquebec.ca/node/107)). All organ procurements were performed after consent was obtained from donors (living donation) or next of kin. In Canada, organ procurement is managed by provincial organizations. Appropriate donor consent was obtained by trained personnel from Transplant Québec (www.transplantquebec.ca/en), our provincial organ procurement organization, using a standardized consent form (S1 Appendix). No vulnerable population was part of the graft procurement pool unless informed consent could be freely obtained by the family or a legal representative, as per national regulations. Donation after circulatory death may only be conducted after palliative care in the Province of Québec. Since the Province of Quebec has a universal public healthcare system, no financial reimbursement was provided to donors' families.

## Study participants

We included all consecutive adults who received a LT from July 1st, 2008 to December 31st, 2017. We did not apply any exclusion criteria.

## Exposures

Our main exposure variable was the intraoperative fluid balance. We defined fluid balance as the total volume of fluid and blood products received, including cellsaver output transfused, minus the diuresis, removed ascites and the total volume of bleeding measured in surgical suction canisters. Volume of administered fluid during a surgery has been widely studied through different fluid strategy protocols in major surgery and is an exposure that can be controlled within a clinical trial [26]. However, since LT is a surgery associated with major bleeding, in contrast to most other major abdominal surgeries, fluid balance might be a better exposure to evaluate [27,28].

Our secondary exposure variable was the presence of vasopressors on ICU admission. We chose this exposure at the end of LT to better reflect the hemodynamic instability associated with post reperfusion syndrome, a potential perioperative factor in graft ischemic injury, and to represent the effect of the overall intraoperative fluid management strategy [9,29,30]. This second exposure was also explored as a potential effect modifier. We included any vasopressor infusion (norepinephrine, vasopressin, phenylephrine, epinephrine) to classify patients as receiving vasopressor at ICU admission. For description purposes and sensitivity analyses, we converted all vasopressors doses to norepinephrine equivalent in ug per kilogram per minute and combined them together to create a single total vasopressor dose variable (vasopressin was converted from units per minute to ug per kilogram per minute in a 1:2.5 ratio and phenylephrine was converted in a 10:1 ratio) [31–33].

## Outcomes

**Primary outcome.** Our primary outcome was a composite outcome of 1-year graft biliary complications (anastomotic stenosis (AS) and non-anastomotic biliary strictures (NAS) (NAS may also be called "ischemic cholangitis" or "post-transplantation cholangiopathy")). We selected biliary strictures as our primary outcome because of their overall high incidence following liver transplantation and their theoretical association with ischemic injury [3,8,9,34].

**Secondary outcomes.** Our secondary outcomes were NAS alone, AS alone, graft hepatic artery complications, primary graft non-function (PNF) and survival up to 1 year. We also measured all vascular graft complications for descriptive purposes.

## Covariables

We included patient's demographic characteristics, preoperative severity of liver failure (MELD-Na), indication for transplantation (including sclerosing cholangitis and acute liver failure), retransplantation status, preoperative hemoglobin level and renal failure status to describe the sample and analyze baseline confounding. We included other baseline characteristics for descriptive purposes. Baseline intraoperative central venous pressure (CVP), the use of an intraoperative phlebotomy [35], length of vena cava clamping and type of vena cava clamping as well as the type of biliary anastomosis were also analyzed as potential confounders. We used the Donor Risk Index (DRI) [23] to represent donors' characteristics as potential confounders. The donor risk index includes the following donor and graft variables: age, height, cause of death, type of graft (NDD, DCD, split), procurement localization (national, regional or local) and cold ischemia time.

## Data sources and measurement

Baseline characteristics, exposures and most covariables were already available in a database used for previous publications (532 transplantations) [36–38]. We extracted data for the previously excluded patients (36 transplantations who needed preoperative RRT) from their medical records and new data specific to this study's objectives for all patients. The present study is thus an extension of our previous ones that both presents new outcomes and includes previously excluded patients. Data was collected retrospectively from patients' charts between September 2018 and March 2020 and could not be fully anonymized. However, donors' data was fully anonymized in recipients' charts. The final data set is stored in a secured server at the Centre de recherche du CHUM and is de-nominalized.

Biliary stricture complications, vascular complications and PNF were extracted from radiological reports and patients' chart. We defined AS as any significant reduction of the main biliary duct anastomosis and NAS as biliary strictures occurring in the intrahepatic or extrahepatic system outside anastomotic site, as judged by the endoscopist on endoscopic retrograde cholangio-pancreatography report or by the radiologist on either magnetic resonance cholangio-pancreatography report or percutaneous cholangiography [39,40]. We defined hepatic artery thrombosis (HAT) as reported in any angiographic diagnostic procedure (computed tomography angiography (CTA) or direct catheter angiography), either on the main artery or any intrahepatic branch. We defined hepatic artery stenosis (HAS) as any of the same angiographic diagnostic procedure describing a stenosis in association with either a resistance index < 0.5 on Doppler ultrasound or a therapeutic intervention within a month of the angiographic imaging (percutaneous or surgical). We defined portal vein and hepatic veins thrombosis as reported by the radiologist on CTA. Every biliary or vascular complication was extracted and classified by one investigator (JL) and then adjudicated by an expert committee (JL, JMG, FMC) to ensure they fit predefined definitions. Finally, we defined PNF as graft failure leading to retransplantation or death, as reported by the liver transplant specialist in patients' chart. Survival was computed as a time from LT to death or censorship (1-year post-LT or retransplantation). The same outcome was already reported in our previous papers on 532 LTs [37,38]. However, we now have included 36 LTs with preoperative renal replacement therapy that were excluded in our previous studies.

MELD was calculated using the preoperative laboratory values closest to transplantation. Donors' characteristics were extracted from a local clinical database and were used to calculate the Donor Risk Index (DRI). The DRI includes age, height, cause of death, DCD status, split liver, local or national procurement as well as cold ischemic time [23]. Both in situ split surgery for pediatric transplantation and living donation were classified as a split liver. Any procurement in a Montreal center was considered local and any procurement outside Montreal was considered national. Donor ethnicity was not available in our data set. Since 90% of the population in the province of Quebec is Caucasian, we considered all donors as such. We included all types of strokes as cerebrovascular accidents (ischemic stroke, hemorrhagic stroke, and subarachnoid hemorrhage).

## Data management

We merged data from existing databases with data extracted and classified from patients' chart and radiological reports in a new electronic database. Collected information was codified and remained strictly confidential.

## Sample size

We postulated a hazard ratio of 1.25 for biliary complications to calculate our sample size. Assuming a proportion of biliary complications of 25% at one year [9], an absolute reduction of 6.25% [27] might be converted to a hazard ratio of 1.4. The selected hazard ratio of 1.25 was thus more conservative. From our preliminary data [37], we knew that the standard deviation of the fluid balance is 2.02 L, that the coefficient of determination of the relationship between exposure and covariables was 0.1 for fluid received and 0.01 for the fluid balance. Thus, 307 patients were required (HR = 1.25, alpha = 0.025, beta = 0.9, proportion of events = 0.25, $R^2$ = 0.2). To increase the power of our secondary analyses, we did a Bonferroni correction for fourteen analyses (sample size of 436) and inflated the sample size by 15% to compensate for potential missing data from the main exposure or outcome (sample size of 501). A cohort of 568 patients was available and we thus included all patients.

## Data analyses

We reported categorical data as proportions and continuous data as means or medians depending on distribution. Our primary analysis was a measure of association between fluid balance and time to biliary complications. We fitted a multivariable Fine and Gray marginal model, adjusted for potential confounders, using robust sandwich standard errors to manage correlation between patients who had more than one transplant in the sample. Death unrelated to biliary complications was considered as a competing risk. Patients were censored at retransplantation (if they received a second transplantation for another cause than a biliary complication within 1 year) or at 365 days. We evaluated the hazard proportionality assumption by the Harrell and Lee test as well as by a visual inspection of the Schoenfeld residuals. We evaluated the linearity of the association between fluid balance and outcome by a visual inspection of the Martingale residuals and by fitting the models with restricted cubic splines and using likelihood ratio tests. The presence of vasopressors was analysed as a potential effect modifier using statistical interaction. If no effect modifying was observed, the presence of vasopressor was analysed as a potential confounder. Multicollinearity was explored using the VIF statistic.

Secondary analyses were the association between fluid balance and non-anastomotic strictures alone, anastomotic strictures alone, hepatic artery complications, primary graft nonfunction and survival. We fitted similar multivariable Fine and Gray marginal models for the first two outcomes. Since PNF was rare and occurred within days to weeks of LT, we used a

logistic regression model fitted by generalized estimating equations (GEE) using an exchangeable correlation matrix and explored non-linearity by a polynomial model. We fitted a marginal Cox model for survival. Since the proportionality hazard assumption was not met, we fitted a time-dependant linear coefficient for fluid balance using a time-transform functionality based on the visual inspection of the Schoenfeld residuals. Due to a lower number of events in our secondary analyses, we limited covariables based on clinical judgment.

We fitted all models in 5 data sets with multiple imputations by chained equations to manage missing values and pooled fitted estimates and variances using the Rubin rule. We conducted sensitivity analyses by fitting all our models on complete cases only. For our primary analysis, we conducted three other sensitivity analyses in data sets with multiple imputations: we considered ascites as a covariable (instead of being with the fluid balance exposure), we used only graft complications occurring in the first three months after transplantation and we used vasopressor doses in norepinephrine equivalent as a covariable (instead of a dichotomic one). Finally, we conducted a post-hoc analysis by exploring the association between hepatic artery complications and biliary complications using logistic regressions fitted by GEE using an exchangeable correlation matrix. We used R software (R foundation, version 3.6.2) to conduct all analyses and reported all analyses with 95% robust confidence intervals (CI).

## Results

We included 568 consecutive liver transplantations performed in 519 patients (S1 Fig in S2 Appendix), of which 98% where donations after neurological determination of death (NDD). Six patients died in the operating room, so 562 liver transplantations done in 514 patients were included in the analyses (1 out of 6 patients who died intraoperatively had a first liver transplantation that was included in our analyses). Baseline characteristics and intraoperative variables are reported according to the presence or absence of any biliary complication (Table 1). Recipient characteristics were similar between the two groups, apart from male patients who seemed to have fewer biliary complications (60% versus 70%). Donor characteristics were also similar between the two groups, with a mean cold ischemic time of 7.4 hours for the whole cohort.

Graft complications up to 1 year after transplantation are reported in Table 2 and S2 Table in S2 Appendix. Amongst our cohort, 192 (34.2%) patients presented either an anastomotic stricture (AS) or a non-anastomotic stricture (NAS) within the first year after transplantation. Anastomotic strictures (n = 167, 29.7%) were more common than non-anastomotic strictures (n = 56, 10.0%). The median time [q1, q3] to diagnosis was 27 days [14, 65] for any biliary complication, 23 days [13, 61] for anastomotic strictures and 59 days [38, 138] for non-anastomotic strictures among patients who had the complication (not in tables). Eighteen patients (3.2%) suffered from a PNF. One-year mortality was 7.6% in the whole cohort (39 patients out of 514 died).

Mean fluid balance and proportion of patients with any vasopressor use are reported in Table 3 and S1 Table in S2 Appendix. Distribution of fluid balance and vasopressors are reported in S3 Fig in S2 Appendix. Distribution of fluid balance or vasopressor doses according to the year of transplantation or to the presence of biliary complications are reported in S4-S6 Figs in S2 Appendix. Amongst patients who received any vasopressor infusion, the median [q1, q3] dose of vasopressors in norepinephrine equivalent was 0.20 ug/kg/min [0.11, 0.37] in the whole cohort, 0.21 ug/kg/min [0.11, 0.38] in patients without biliary complications and 0.19 ug/kg/min [0.10, 0.36] in patients with biliary complications (S6 Fig in S2 Appendix).

Our main analysis is reported in Table 4. We did not observe any effect of fluid balance on biliary complications (HR = 0.97; 95% CI, 0.93 to 1.02). The presence of vasopressors on ICU

**Table 1. Cohort characteristics.**

| | No biliary complication (n = 370) | Biliary complication (n = 192) | Overall[1] (n = 568) |
|---|---|---|---|
| **Recipient characteristics** | | | |
| Age | 56 [45,61] | 55 [47, 61] | 55 [46,61] |
| Sex (male) | 260 (70%) | 116 (60%) | 379 (67%) |
| Hemoglobin level (g/L) | 105 (24) | 106 (24) | 105 (24) |
| Platelet count (x 10$^9$/L) | 83 [52, 123] | 80 [49, 118] | 83 [51, 122] |
| Bilirubin level (umol/L) | 69 [34, 159] | 66 [27, 183] | 69 [32, 170] |
| Creatinine level (umol/L) | 81 [63, 110] | 79 [62, 108] | 81 [63, 110] |
| INR | 1.6 [1.3, 2.0] | 1.6 [1.3, 2.0] | 1.6 [1.3, 2.0] |
| Fibrinogen (g/L) | 2.1 [1.3, 2.9] | 2.1 [1.3, 3.0] | 2.1 [1.2, 2.9] |
| Sodium (mmol/L) | 136 (5) | 136 (5) | 136 (5) |
| MELD | 21 (9) | 21 (9) | 21 (9) |
| Preoperative renal replacement therapy | 17 (4.6%) | 14 (7.3%) | 32 (5.6%) |
| Indication of transplantation | | | |
| • Cirrhosis | 286 (77%) | 154 (80%) | 445 (78%) |
| • ALF | 18 (5%) | 7 (4%) | 24 (4%) |
| • Other | 21 (6%) | 14 (7%) | 36 (6%) |
| • Retransplantation | 45 (12%) | 17 (9%) | 63 (11%) |
| HCC | 85 (23%) | 49 (26%) | 135 (24%) |
| Transplantation for an autoimmune disorder | 82 (22%) | 41 (21%) | 124 (22%) |
| Transplantation for sclerosing cholangitis | 37 (10%) | 12 (6%) | 50 (9%) |
| **Donor characteristics** | | | |
| Age | 54 [38, 64] | 56 [45, 65] | 55 [41, 64] |
| Sex (male)[2] | 215 (58%) | 99 (52%) | 316 (56%) |
| CIT (hours) | 7.4 (2.3) | 7.3 (2.4) | 7.4 (2.3) |
| Type of donation: | | | |
| Donation after NDD | 363 (98%) | 557 (98%) | 557 (98%) |
| • DCD | 1 (< 1%) | 1 (< 1%) | 2 (< 1%) |
| • Living | 6 (2%) | 2 (2%) | 9 (2%) |
| Cause of death (excludes living donation) | | | |
| Anoxia | 71 (19%) | 30 (16%) | 102 (18%) |
| • Hemorrhagic stroke | 53 (14%) | 29 (15%) | 82 (14%) |
| • Ischemic stroke | 118 (32%) | 73 (38%) | 193 (34%) |
| • SAH | 24 (6%) | 15 (8%) | 39 (7%) |
| • Traumatic brain injury | 84 (22%) | 35 (18%) | 119 (21%) |
| • Other | 8 (2%) | 4 (2%) | 12 (2%) |
| • NDD–cause not reported | 6 (2%) | 3 (2%) | 12 (2%) |
| DRI[3] | 2.2 [1.7, 2.6] | 2.2 [1.9, 2.7] | 2.2 [1.8, 2.7] |
| **Surgical variables** | | | |
| Vena cava clamping time (minutes) | 39 [33, 46] | 40 [35, 46] | 40 [33, 46] |
| Length of surgery (minutes) | 240 [205, 275] | 240 [200, 289] | 245 [203, 280] |
| Piggyback vena cava anastomosis | 20 (5.4%) | 10 (5.2%) | 30 (5.3%) |
| Hepatico-jejunostomy | 101 (27%) | 32 (17%) | 133 (24%) |
| **Bleeding, fluid and hemodynamic variables** | | | |
| Baseline CVP (mmHg) | 14 (5) | 15 (5) | 14 (5) |
| Phlebotomy (exposed) | 200 (54%) | 104 (54%) | 306 (54%) |
| Ascites (L)[4] | 1.2 [0, 5.0] | 1.5 [0, 5.0] | 1.3 [0, 5.0] |
| Intraoperative urine output (L) | 0.3 [0.2, 0.5] | 0.4 [0.2, 0.5] | 0.3 [0.2, 0.5] |
| Intraoperative bleeding (L) | 1.0 [0.6, 2.0] | 1.0 [0.7, 1.9] | 1.0 [0.6, 2.0] |
| Crystalloid (L) | 4.0 [3.0, 4.8] | 4.0 [3.0, 5.0] | 4.0 [3.0, 5.0] |
| Synthetic colloid (L) | 0 [0, 0.5] | 0 [0, 0.5] | 0 [0, 0.5] |

(*Continued*)

**Table 1.** (Continued)

| | No biliary complication (n = 370) | Biliary complication (n = 192) | Overall[1] (n = 568) |
|---|---|---|---|
| Iso-oncotic albumin (L) | 0 [0, 0] | 0 [0, 0.5] | 0 [0, 0] |
| Cellsaver output (L) | 0.2 [0.1, 0.4] | 0.3 [0.2, 0.5] | 0.3 [0.1, 0.5] |
| Any intraoperative RBC transfusion | 102 (28%) | 57 (30%) | 161 (28%) |
| Any intraoperative labile blood product transfusion other than RBC | 59 (16%) | 32 (17%) | 91 (16%) |

Results are reported as number of observed cases (proportion in %), as means (SD) or as medians [quartile 1, quartile 3] according to the shape of the distribution.

[1] Overall cohort descriptive data included 6 intraoperative deaths that are excluded from all analyses. Data from the 6 patients excluded from the analyses are only presented in Table 1.

[2] 10 missing values in the "no complication group" and 3 missing values in the "complication" group.

[3] 10 missing values in the "no complication group" and 5 missing values in the "complication" group.

[4] 70 missing values in the "no complication group" and 28 missing values in the "complication" group.

Abbreviations: INR = International Normalized ratio, MELD = Model for End–stage Liver Disease, ALF = Acute Liver Failure, HCC = Hepatocellular Carcinoma, CIT = Cold Ischemia Time, NDD = Neurological Determination of Death, DCD = Donation after Circulatory Death, SAH = Subarachnoid Hemorrhage, DRI = Donor Risk Index, CVP = Centre Venous Pressure, RBC = Red Blood Cells, ICU = intensive care unit.

admission was also not associated with more biliary complications (HR, 1.04; 95% CI, 0.76 to 1.43). Statistical interaction was not significant between vasopressor use and fluid balance. In multivariable analyses, we observed that male patients had a lower risk of developing biliary complications (HR, 0.63; 95% CI, 0.46 to 0.87). Our analysis was robust to complete cases analysis and other sensitivity analyses (Table 4 and S2-S4 Tables in S2 Appendix).

Our secondary analyses are reported in Table 5 and S5-S8 Tables in S2 Appendix. We did not observe any effect of fluid balance on non-anastomotic strictures, anastomotic strictures or hepatic artery complications (S5-S7 Tables in S2 Appendix). However, we observed a non-linear association between fluid balance and primary graft non-function, with a higher fluid balance being associated exponentially with a higher risk of PNF (S8 Table and S7 Fig in S2 Appendix). We observed that a positive fluid balance decreased survival (HR = 1.40, 95% CI, 1.14 to 1.71) (Table 5 et S8 Fig in S2 Appendix). This effect linearly decreased over

**Table 2. Graft complications up to 1 year after transplantation.**

| Graft complications | Overall (n = 562) |
|---|---|
| **Biliary complications (any)[1]** | **192 (34.2%)** |
| Non-anastomotic stricture | 56 (10.0%) |
| Anastomotic stricture | 167 (29.7%) |
| Both non-anastomotic and anastomotic strictures | 31 (5.5%) |
| **Vascular complications (any)** | **121 (21.5%)** |
| Hepatic artery stenosis | 73 (13.0%) |
| Hepatic artery thrombosis | 33 (5.9%) |
| Portal vein thrombosis | 38 (6.8%) |
| Hepatic vein thrombosis | 12 (2.1%) |
| **Primary graft non-function[2]** | **18 (3.2%)** |
| **Retransplantation (any indication)[3]** | **35 (6.2%)** |

Results are reported as number of observed cases (proportion in %).

[1] 154 complications out of 192 occurred (80%) within 3 months of transplantation.

[2] Includes some patients with a vascular complication.

[3] Includes patients with biliary or vascular complications, primary non–function and any other cause.

**Table 3. Fluid balance and vasopressor use.**

| | Overall (n = 562) | No biliary complication (n = 370) | Biliary complication (n = 192) |
|---|---|---|---|
| Fluid balance including ascites(L)[1] | 0.7 (3.7) | 0.7 (3.6) | 0.6 (3.8) |
| Fluid balance excluding ascites (L)[2] | 3.5 (1.7) | 3.5 (1.7) | 3.4 (1.9) |
| Any vasopressor upon ICU admission | 269 (48%) | 173 (47%) | 96 (50%) |
| Norepinephrine infusion upon ICU admission[3] | 255 (45%) | 164 (44%) | 91 (47%) |
| Vasopressin infusion upon ICU admission | 160 (28%) | 109 (29%) | 51 (27%) |
| Any inotrope upon ICU admission | 3 (1%) | 3 (1%) | 0 (0%) |

Results are reported as number of observed cases (proportion in %) or as means (SD).

[1] 74 missing values in the "no complication group" and 30 missing values in the "complication" group.

[2] 4 missing values in the "no complication group" and 2 missing values in the "complication" group.

Abbreviations: ICU = Intensive Care Unit.

postoperative time, with late events being less associated with intraoperative fluid balance (Table 5 and S9 Fig in S2 Appendix). We observed no interaction between vasopressor and fluid balance on any outcome and our results were robust to complete cases analyses.

As a post-hoc analysis, we explored the cross-sectional association between any hepatic artery complication and biliary complications (Table 6). Hepatic artery stenosis was reported in 10% of patients (37 out of 370) with no biliary complications and 19% of patients with any biliary complication (36 out of 192; 26 out of 167 with AS (16%) and 17 patients out of 56 with

**Table 4. Any biliary complication.**

| Variable | HR With imputations (n = 562) | HR Complete cases (n = 442) |
|---|---|---|
| Fluid balance (L) | 0.97 [0.93, 1.02] | 0.99 [0.95, 1.04] |
| Any vasopressor | 1.04 [0.76, 1.43] | 1.03 [0.72, 1.47] |
| Phlebotomy | 0.94 [0.65, 1.37] | 0.95 [0.64, 1.42] |
| Age (10 years) | 0.95 [0.82, 1.09] | 0.93 [0.80, 1.08] |
| Sex (male) | 0.63 [0.46, 0.87]* | 0.61 [0.43, 0.87]* |
| MELD | 0.99 [0.97, 1.01] | 1.00 [0.97, 1.03] |
| Sclerosing cholangitis | 1.11 [0.53, 2.33] | 1.09 [0.44, 2.68] |
| ALF | 0.66 [0.29, 1.54] | 0.46 [0.16, 1.34] |
| Retransplantation | 1.14 [0.58, 2.26] | 1.21 [0.52, 2.78] |
| Preoperative RRT | 1.75 [0.95, 3.23] | 1.55 [0.81, 2.94] |
| Preoperative creatinine (10 umol/L) | 0.99 [0.97, 1.01] | 0.99 [0.97, 1.02] |
| Preoperative hemoglobin (10 g/L) | 1.06 [0.97, 1.15] | 1.08 [0.98, 1.18] |
| Baseline CVP (mmHg) | 1.03 [0.99, 1.06] | 1.02 [0.99, 1.05] |
| Piggyback | 1.23 [0.65, 2.35] | 1.36 [0.70, 2.62] |
| Vena cava clamping time (10 min) | 1.08 [0.97, 1.19] | 1.07 [0.97, 1.19] |
| Hepaticojejunostomy | 0.55 [0.32, 0.96]* | 0.48 [0.24, 0.95]* |
| DRI | 1.24 [0.99, 1.56] | 1.15 [0.90, 1.48] |

* Statistically significant.

Results from a model fitted on data sets with multiple imputations.

Results are expressed with 95% confidence intervals.

Statistical interaction between fluid balance and presence of vasopressors at ICU admission was not significant in either model.

Abbreviations: HR = Hazard Ratio, MELD = Model for End–Stage Liver Disease, ALF = Acute Liver Failure, RRT = Renal Replacement Therapy, DRI = Donor Risk Index, ICU = Intensive Care Unit.

**Table 5. Survival up to 1 year.**

| Variable | HR With imputations (n = 562) | HR Complete cases (n = 442) |
|---|---|---|
| Fluid balance (L) | 1.40 [1.14, 1.71]* | 1.36 [1.08, 1.72]* |
| Fluid balance (L) * time (days)ᵃ | 0.99 [0.99, 0.99]* | 0.99 [0.99, 0.99]* |
| Any vasopressor | 0.85 [0.41, 1.74] | 0.89 [0.39, 2.03] |
| Phlebotomy | 0.44 [0.20, 0.96]* | 0.51 [0.22, 1.22] |
| Age (10 years) | 1.62 [1.18, 2.22]* | 1.66 [1.14, 2.42]* |
| Sex (male) | 1.14 [0.57, 2.27] | 0.91 [0.42, 1.96] |
| MELD | 1.04 [1.01, 1.08]* | 1.04 [0.99, 1.09] |
| Retransplantation | 2.30 [1.03, 5.15]* | 3.96 [1.65, 9.52]* |
| Preoperative RRT | 0.71 [0.21, 2.37] | 0.64 [0.15, 2.71] |
| DRI | 1.33 [0.88, 2.01] | 1.36 [0.85, 2.19] |

\* Statistically significant.

Results of a model fitted on data sets with multiple imputations, stratified for ALF.

Results are expressed with 95% confidence intervals.

Statistical interaction between fluid balance and presence of vasopressors at ICU admission was not significant in either model.

A HR > 1 increases the risk of death.

1–year mortality was 7.6% in the whole cohort (39 patients out of 514 died), 8.4% in patients without biliary complications (29 patients out of 347 died) and 5.9% in patients with biliary complications (10 patients out of 167 died). 6 intraoperative death were excluded; the real 1–year mortality of our full cohort is thus 8.7% (45 out of 519 patients).

ᵃ Fluid balance had a decreasing effect on survival over time, which means that late events were less associated with intraoperative fluid balance. HR for fluid balance was linearly time dependent and the HR for fluid balance and time interaction was 0.9989 (p = 0.009). The overall effect of fluid balance in the model with imputations was thus: $HR = e^{0.325*\text{fluid balance}- 0.001*\text{fluid balance}*\text{days}}$.

Abbreviations: HR = Hazard Ratio, MELD = Model for End–Stage Liver Disease, RRT = Renal Replacement Therapy, DRI = Donor Risk Index, ALF = Acute Liver Failure, ICU = Intensive Care Unit.

NAS (30%)). Such stenosis was strongly associated with any biliary complication (OR, 2.10; 95% CI, 1.21 to 3.66) and had a stronger association with NAS (OR, 3.70; 95% CI, 1.87 to 7.32).

## Discussion

In this retrospective cohort study of 568 consecutive liver transplantations, we observed that 34% of recipients had a biliary complication 1 year after transplantation. We did not observe any effect of fluid balance or use of vasopressors on ICU admission on these complications but observed a harmful effect of a higher fluid balance on primary graft non-function and survival. Hepatic artery stenosis was also strongly associated with biliary complications.

Biliary complications are believed to be at least partly mediated through ischemic and ischemia-reperfusion injury or immune mechanisms [9,17]. In this study, we did not observe any effect of fluid balance or vasopressors on such complications. Fluid balance is probably a non-specific exposure with intrinsic limitations in the liver transplantation context when the underlying intraoperative hemodynamic instability and the total vasopressor exposure are not measured. Although such instability might have been a better exposure of interest to explore the effects of intraoperative variables on biliary complications, it would not have considered the hemodynamic management strategy applied by anesthesiologists. One recent pragmatic randomized trial found no difference between a restrictive and a liberal fluid management

**Table 6. Cross–sectional association between hepatic artery complications and biliary complications.**

| | n (%) | OR |
|---|---|---|
| **Biliary complications (N = 192)** | | |
| Hepatic artery stenosis | 36 (18.8%) | 2.10 [1.21, 3.66]* |
| Hepatic artery thrombosis | 12 (6.3%) | 0.73 [0.25, 2.15] |
| Both hepatic artery stenosis and thrombosis | 8 (4.2%) | 1.90 [0.71, 5.14] |
| Either hepatic artery stenosis or thrombosis | 40 (20.9%) | – |
| **Non-anastomotic strictures (N = 56)** | | |
| Hepatic artery stenosis | 17 (30.4%) | 3.70 [1.87, 7.32]* |
| Hepatic artery thrombosis | 7 (12.5%) | 2.86 [0.84, 9.75] |
| Both hepatic artery stenosis and thrombosis | 4 (7.1%) | 3.68 [1.11, 12.15]* |
| Either hepatic artery stenosis or thrombosis | 20 (35.7%) | – |
| **Anastomotic strictures (N = 167)** | | |
| Hepatic artery stenosis | 26 (15.6%) | 1.35 [0.75, 2.41] |
| Hepatic artery thrombosis | 9 (5.4%) | 0.49 [0.13, 1.80] |
| Both hepatic artery stenosis and thrombosis | 6 (3.6%) | 1.34 [0.47, 3.77] |
| Either hepatic artery stenosis or thrombosis | 29 (17.4%) | – |
| **No biliary complications (N = 370)** | | |
| Hepatic artery stenosis | 37 (10.0%) | – |
| Hepatic artery thrombosis | 21 (5.7%) | – |
| Both hepatic artery stenosis and thrombosis | 9 (2.4%) | – |
| Either hepatic artery stenosis or thrombosis | 49 (13.2%) | – |

* Statistically significant.

OR are expressed with 95% confidence intervals and were fitted on 562 observations.

Since any biliary complication and anastomotic strictures are frequent events, the OR overestimates the true risk.

Abbreviations: OR = Odds Ratio.

strategy on 1-year disabilities after major abdominal surgeries but reported a 3.5% increase in 30-day acute kidney injury in the restrictive group [41]. However, the liver transplantation population has different physiological characteristics and undergoes much higher blood loss. Since inference from the major surgery population is limited, better data on fluid management in liver transplantation should be gathered [42].

The association between hepatic artery thrombosis and biliary complications is well described [9]. We did not exclude patients with hepatic artery thrombosis or stenosis in our analyses since arterial complications may be a mediator in the causal relationship between intraoperative hemodynamic management and biliary complications. Indeed, we observed an association between hepatic artery stenosis and biliary complications. Such association was stronger with non-anastomotic strictures (OR, 3.70; 95% CI, 1.87 to 7.32), supporting that observed non-anastomotic strictures may partly originate from reduced hepatic arterial flow. Such observation may support that higher quality hemodynamic variables, as surrogates of graft perfusion, combined with fluid management strategy should be further studied in a prospective fashion to better define their effect on graft outcomes.

We also observed that male patients were less likely to develop biliary complications (HR, 0.64; 95% CI, 0.46 to 0.88) and such effect was explained by a lower incidence of anastomotic strictures (HR = 0.64; 95% CI, 0.46 to 0.90). This biological sex effect may further support an immune-mediated biliary injury mechanism for biliary complications. A prospective randomized trial of immunosuppressive drug withdrawal found male gender to be an independent predictor of successful immunosuppression weaning, suggesting that a higher

immunosuppressive regimen was required in female liver transplant recipients [43]. A higher immune response in women may thus explain their higher risk of biliary complications. Other outcomes in liver transplant recipients are different according to biological sex and our results confirm that such effects should not be overlooked [44,45].

In our previous papers, we did not observe any association between fluid balance or vasopressor use at ICU admission and postoperative acute kidney injury, but we observed some effects of a higher fluid balance on the need for renal replacement therapy [37,38]. In the present study, we did not observe any association between the same exposure variables and biliary graft complications, but we observed an association of our exposure variables with more PNF and a lower survival. In the present survival analysis, we included 7% more patients who needed preoperative renal replacement therapy (who were previously excluded) and confirmed the effect of fluid balance on lower survival even with sicker patients included in the analysis [37,38]. Our exposure thus seems to be associated with more severe outcomes, suggesting either that there is no clinical effect on less severe outcome or that our exposure measurement is biased by non-differential errors and may thus be significantly associated only with stronger effects.

One of the main strengths of this study is the high number of consecutive patients included and the absence of lost to follow-up, precluding selection bias. It is the largest cohort study to explore the effect of intraoperative variables on liver graft complications. Although this study was retrospective, the main exposure of interest (fluid balance) had been collected prospectively in a local database. Also, our outcome classification has a low probability of bias. All biliary and vascular complications were collected directly from imaging or procedure reports and adjudicated by an expert committee blinded to the exposure. This rigorous outcome classification might explain our high rate of biliary strictures. Our analyses were well-adapted to the data structure and we conducted multiple sensitivity analyses that confirmed the robustness of our findings.

## Limitations

This study has many limitations that must be acknowledged. Our results have limited external validity since they are based on data from only one liver transplantation center. Our perioperative management, such as the use of phlebotomy in 54% of our transplantations and our low transfusion needs might differ from other centers. Also, the vast majority of donations were after NDD and most grafts were implanted with a vena cava replacement technique, which may limit transportability to other populations. There is always a risk for non-differential classification bias if any measurement error occurred for either the exposure or the outcome. Our fluid balance exposure is a composite variable, which includes the total volume of bleeding in surgical suctions; we could not know the volume of ascites included in this measure. Such non-differential misclassification would bias our hypothesis toward the null effect, reducing any observable association. Our primary outcome was a composite of all biliary complications, but mechanisms that lead to anastomotic strictures may be different that those leading to non-anastomotic strictures. However, we explored such differential effects through sensitivity analyses.

Many cofounders could influence the association between fluid balance or vasopressors and graft complications. We adjusted our statistical models for selected covariables with potential association with both the exposures and the outcomes. However, such adjustment was limited for rarer outcome, such as PNF. Also, as previously mentioned, overall hypotension exposure, acute changes in hemodynamics and its severity, rapid bleeding, pre-emptive actions based on surgical manoeuvres and the clinical feeling of anaesthesiologists were not

captured by our study design. A malfunctioning graft may also be associated with a higher need for fluid and vasopressors after reperfusion, with potential reverse causation caused by residual confounding (insufficient adjustment by the DRI) for the PNF outcome. Uncontrolled and residual confounding are thus the major limitations of this study.

## Conclusion

This study is the first to report the effect of fluid balance on biliary graft complications in liver transplant recipients and to our knowledge, the largest study to report the effect of intraoperative variables on graft complications. We observed that fluid balance was not associated with biliary complications in liver transplant recipients. Intraoperative hemodynamic variables, including administered fluid volume and vasopressors, should be further investigated to explore the effect of intraoperative hemodynamic management on liver graft complications and help improve graft survival.

## Supporting information

**S1 Appendix. Donor consent form in the Province of Quebec.**
(PDF)

**S2 Appendix. Supplemental tables and figures.**
(DOCX)

**S3 Appendix. STROBE checklist.**
(PDF)

## Acknowledgments

We would like to thank Kip Brown from the Centre de recherche du CHUM for his help in collecting some data as well as all anesthesiologists from the liver transplantation team who helped collected prospective data over the past decade.

## Author Contributions

**Conceptualization:** Jordan Larivière, Jeanne-Marie Giard, Michaël Chassé, François Martin Carrier.

**Data curation:** Jordan Larivière, Jeanne-Marie Giard, Rui Min Zuo, Luc Massicotte, François Martin Carrier.

**Formal analysis:** François Martin Carrier.

**Funding acquisition:** François Martin Carrier.

**Investigation:** Jordan Larivière, Jeanne-Marie Giard, François Martin Carrier.

**Methodology:** Jordan Larivière, Michaël Chassé, François Martin Carrier.

**Project administration:** François Martin Carrier.

**Supervision:** François Martin Carrier.

**Validation:** François Martin Carrier.

**Writing – original draft:** Jordan Larivière, François Martin Carrier.

**Writing – review & editing:** Jordan Larivière, Jeanne-Marie Giard, Rui Min Zuo, Luc Massicotte, Michaël Chassé, François Martin Carrier.

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
