## [Decision Letter · Decision Letter 0]

16 Jun 2021

PONE-D-21-17298

Association between intraoperative fluid balance, vasopressors and graft complications in liver transplantation: a cohort study

PLOS ONE

Dear Dr. Carrier,

Thank you for submitting your manuscript to PLOS ONE. After careful consideration, we feel that it has merit but does not fully meet PLOS ONE’s publication criteria as it currently stands. Therefore, we invite you to submit a revised version of the manuscript that addresses the points raised during the review process.  The study has merit and we encourage a resubmission.

Please submit your revised manuscript within 60 days. If you will need more time than this to complete your revisions, please reply to this message or contact the journal office at plosone@plos.org. Please include the following items when submitting your revised manuscript:

We look forward to receiving your revised manuscript.

Kind regards,

Gianfranco D. Alpini

Academic Editor

PLOS ONE

Journal Requirements:

2.  In ethics statement in the manuscript and in the online submission form, please provide additional information about the patient records/samples used in your retrospective study. Specifically, please ensure that you have discussed whether all data/samples were fully anonymized before you accessed them and/or whether the IRB or ethics committee waived the requirement for informed consent. If patients provided informed written consent to have data/samples from their medical records used in research, please include this information.

3. We note that your study involved tissue/organ transplantation. Please provide the following information regarding tissue/organ donors for transplantation cases analyzed in your study.  

1. Please provide the source(s) of the transplanted tissue/organs used in the study, including the institution name and a non-identifying description of the donor(s).

2. Please state in your response letter and ethics statement whether the transplant cases for this study involved any vulnerable populations; for example, tissue/organs from prisoners, subjects with reduced mental capacity due to illness or age, or minors.

- If a vulnerable population was used, please describe the population, justify the decision to use tissue/organ donations from this group, and clearly describe what measures were taken in the informed consent procedure to assure protection of the vulnerable group and avoid coercion.  

- If a vulnerable population was not used, please state in your ethics statement, “None of the transplant donors was from a vulnerable population and all donors or next of kin provided written informed consent that was freely given.”

3. In the Methods, please provide detailed information about the procedure by which informed consent was obtained from organ/tissue donors or their next of kin. In addition, please provide a blank example of the form used to obtain consent from donors, and an English translation if the original is in a different language.

4. Please indicate whether the donors were previously registered as organ donors. If tissues/organs were obtained from deceased donors or cadavers, please provide details as to the donors’ cause(s) of death.

5. Please provide the participant recruitment dates and the period during which transplant procedures were done (as month and year).

6. Please discuss whether medical costs were covered or other cash payments were provided to the family of the donor. If so, please specify the value of this support (in local currency and equivalent to U.S. dollars).

Additional Editor Comments (if provided):

Reviewers' comments:

Reviewer's Responses to Questions

**Comments to the Author**

1. Is the manuscript technically sound, and do the data support the conclusions?

Reviewer #1: Partly

2. Has the statistical analysis been performed appropriately and rigorously? 

Reviewer #1: I Don't Know

3. Have the authors made all data underlying the findings in their manuscript fully available?

Reviewer #1: Yes

4. Is the manuscript presented in an intelligible fashion and written in standard English?

Reviewer #1: Yes

5. Review Comments to the Author

Reviewer #1: Larivière J and co authors in the current paper examined the possible relationship between intraoperative fluid balance and possible liver complication after liver transplantation. From their data the authors conclude that no relationship is observed between changes in fluid administration/loss and post transplant biliary complication. On the other hand higher fluid balance showed association with survival and PNF.

The paper raises the following comments:

1) A recent study by the same institution examined in a nearly identical patient group fluid balance and acute kidney injury. I was wondering why these data were not included in the previous research for a more complete study also because the main relationship with survival was also previously observed. I think that the authors should clearly report in the introduction or discussion that: i) this represents an extension of previous data ii) data on survival were already reported. Moreover the previous negative results on kidney function should be also described.

2) Anastomotic and non-anastomotic biliary strictures present different risk factors and pathophisiological mechanisms (PMID: 18508368). I do not think they should be pooled together for an adequate analysis. I would suggest to authors to include Figure S5 and the relative statistics. I would put the pooled data in the supplemental materials.

3) I can understand the hypothesis of a relationship between kidney function and fluid balance but the possibility that this may affect biliary strictures seems to me more remote. Since (as stated also by the authors) biliary strictures might be related to ischemic damage, it would have more sense to report blood loss/replacement during surgery beside the fluid balance. Please report these data.

4) The authors reported that ischemia-reperfusion injury is the main cause of non-anastomotic biliary stricture. This does not seem correct. The damage occurring after arterial thrombosis is ischemic only, and the general concept of ischemia-reperfusion refers to what is happening during liver reperfusion. Please change.

5) Criteria for diagnosis of PNF should be reported in methods section and cases of delayed graft function should also be included in statistic as a separate group.

6) There are some inconsistencies between data in the abstract and tables or text. For instance: in abstract (results section) 562 patients are reported; in the text (data source) they seem to be 568 then in the text results section these are actually 562 liver transplants in 514 patients!? Again in the abstract 31 non anastomotic biliary strictures are reported while they seem to be 56 in Table 2 !? Please check with accuracy the reported number in abstract, text and tables. Also in Figure S1: are these transplants or patients?

7) I would include figure s 8 in the main text.

6. PLOS authors have the option to publish the peer review history of their article (what does this mean?). If published, this will include your full peer review and any attached files.

Reviewer #1: No

---

## [Author Response · Author response to Decision Letter 0]

24 Jun 2021

Editor’s comments

We did the required changes. 

2. In ethics statement in the manuscript and in the online submission form, please provide additional information about the patient records/samples used in your retrospective study. Specifically, please ensure that you have discussed whether all data/samples were fully anonymized before you accessed them and/or whether the IRB or ethics committee waived the requirement for informed consent. If patients provided informed written consent to have data/samples from their medical records used in research, please include this information.

We added a sentence that mentioned that the study was conducted with the need for consent waived by our REB (as per our national regulations) in the “Study design and settings” subsection as well as in the online submission form. We clarified the anonymized nature of the data in the “Data sources and measurement” subsection. 

3. We note that your study involved tissue/organ transplantation. Please provide the following information regarding tissue/organ donors for transplantation cases analyzed in your study. 

1. Please provide the source(s) of the transplanted tissue/organs used in the study, including the institution name and a non-identifying description of the donor(s).

We added some required information regarding our organ procurement organization in the “Study design and settings” subsection. Aggregate data on donor characteristics may be found in Table 1. 

2. Please state in your response letter and ethics statement whether the transplant cases for this study involved any vulnerable populations; for example, tissue/organs from prisoners, subjects with reduced mental capacity due to illness or age, or minors.

a. If a vulnerable population was used, please describe the population, justify the decision to use tissue/organ donations from this group, and clearly describe what measures were taken in the informed consent procedure to assure protection of the vulnerable group and avoid coercion. 

b. If a vulnerable population was not used, please state in your ethics statement, “None of the transplant donors was from a vulnerable population and all donors or next of kin provided written informed consent that was freely given.”

We added the required statements in the “Study design and settings” subsection. Since donors’ data was fully anonymized, it is not possible to know what proportion of donor could be perceived as coming from a vulnerable population. In case minors, prisoners, military personnel or patient from any other potentially vulnerable populations was part of the donor pool, informed consent for organ procurement was obtained from the patient, a next of kin or a legal representative as per national regulations. 

3. In the Methods, please provide detailed information about the procedure by which informed consent was obtained from organ/tissue donors or their next of kin. In addition, please provide a blank example of the form used to obtain consent from donors, and an English translation if the original is in a different language.

We added required statements in the “Study design and settings” subsection. A blank example of the consent form was already available in S1 Appendix. 

4. Please indicate whether the donors were previously registered as organ donors. If tissues/organs were obtained from deceased donors or cadavers, please provide details as to the donors’ cause(s) of death.

We added required statements in the “Study design and settings” subsection. Most donors were cadaveric donors. Details as to the donors’ cause(s) of death is detailed in Table 1.

5. Please provide the participant recruitment dates and the period during which transplant procedures were done (as month and year).

We added this information the “Study design and settings” subsection as well as in the “Data sources and management” subsection.

6. Please discuss whether medical costs were covered or other cash payments were provided to the family of the donor. If so, please specify the value of this support (in local currency and equivalent to U.S. dollars).

We added required statements in the “Study design and settings” subsection.

4. Please include captions for your Supporting Information files at the end of your manuscript, and update any in-text citations to match accordingly. Please see our Supporting Information guidelines for more information.

We did the required changes. 

Reviewer’s comments

Reviewer #1 

1) A recent study by the same institution examined in a nearly identical patient group fluid balance and acute kidney injury. I was wondering why these data were not included in the previous research for a more complete study also because the main relationship with survival was also previously observed. I think that the authors should clearly report in the introduction or discussion that: i) this represents an extension of previous data ii) data on survival were already reported. Moreover, the previous negative results on kidney function should be also described.

We did not include the results reported within the present manuscript in previous papers because such outcome data was not collected at the time we conducted the previous studies. Moreover, to conduct this study, we needed to extract new data from patients’ charts and three authors (JL, JMG and FC) adjudicated together all outcomes, a task completed over more than a year (September 2018 to March 2020). The results reported here are genuine and answered a new objective, which was to estimate the association between intraoperative fluid balance/vasopressors and graft outcomes. 

Our previous study published in the same journal was conducted on 532 liver transplantations since we excluded patients with preoperative renal replacement therapy. Since we did not exclude these patients for the present manuscript under review, our sample was slightly larger (please see the “Data sources and management” subsection for detailed clarification) and our survival analysis includes 568 liver transplantations rather than 532. The added patients were at higher risk of death, so the survival analysis was conducted on a slightly different cohort of patient which could have produced different results. Also, the survival analysis included in the present paper was estimated with preoperative need for renal replacement therapy and the donor risk index as a covariable, which was not the case of the survival analysis included in our previous papers. We clarified this in the “Data sources and measurement” subsection along clarifying that the present study was an extension of previous ones. We also added a paragraph in the discussion regarding our previous papers and the fact that 94% of the present cohort was included in the previously reported survival analysis. We also discussed previous results on AKI and possible explanations for our observed results across the three studies. 

2) Anastomotic and non-anastomotic biliary strictures present different risk factors and pathophysiological mechanisms (PMID: 18508368). I do not think they should be pooled together for an adequate analysis. I would suggest to authors to include Figure S5 and the relative statistics. I would put the pooled data in the supplemental materials.

Thank you for this comment. We acknowledge that such complications have many different pathophysiological mechanisms and pathways. However, our hypothesis was that fluid balance and vasopressors could have some effects on both biliary outcomes since ischemic injury could prevent appropriate biliary anastomosis healing (similar to digestive tract anastomosis healing that is affected by intraoperative fluid management) and/or cause non-anastomotic strictures through graft ischemic injury. We further detailed and clarified our hypothesis in the 3rd paragraph of the introduction section. 

Our primary analysis was planned at the protocol stage to be conducted on pooled biliary outcomes with non-anastomotic strictures alone as secondary outcomes. Our sample size was powered for such outcome and our statistical plan was to conduct such analysis as our primary analysis. We decided to analyze non-anastomotic strictures alone as a secondary outcome, since this outcome may be more influenced by our exposure of interest, as insightfully stated by the reviewer. Thus, we believe it is preferable to present our results as such. For consistency, we added a model for anastomotic strictures. 

Figure S5 is a graphical representation of the distribution of fluid balance according to the biliary outcomes. The presented fluid balance is the variable included in our analyses, so the relevant statistics are the one reported with our results from our multivariable analyses (biliary complications, non-anastomotic strictures and anastomotic strictures). Moreover, conducting a statistical test on an exposure variable from an outcome-based categorization is not recommended. 

3) I can understand the hypothesis of a relationship between kidney function and fluid balance but the possibility that this may affect biliary strictures seems to me more remote. Since (as stated also by the authors) biliary strictures might be related to ischemic damage, it would have more sense to report blood loss/replacement during surgery beside the fluid balance. Please report these data.

We agree that our hypothesis may seem “remote”, but we believe it is innovative to explore the potential effects of intraoperative fluid balance and vasopressor need/use on post-transplant graft complications in liver transplantation. To our knowledge, nobody has explored such effect within pre-defined epidemiological models adjusted for potential confounders. As stated in answer to comment #2, we clarified our hypothesis in the introduction section. 

Blood loss and replacement during surgery is reported in Table 1. We changed the name of the sub-heading “Hemodynamic variables” for “Bleeding, fluid and hemodynamic variables” to enhance clarity. Thank you for this comment. 

4) The authors reported that ischemia-reperfusion injury is the main cause of non-anastomotic biliary stricture. This does not seem correct. The damage occurring after arterial thrombosis is ischemic only, and the general concept of ischemia-reperfusion refers to what is happening during liver reperfusion. Please change.

Outside hepatic artery thrombosis or stenosis, we believed it is hard to differentiate between purely ischemic injury and ischemic-reperfusion injury, since perfusion is “re-established” once hemodynamic instability is resolved. However, we do not believe our study is adding any clarification of this injury mechanism. As suggested, we modified sentences across the manuscript to report that NAS may be caused by ischemic injury although we kept the discussion as such about graft ischemic and ischemia-reperfusion injury that occur during liver transplantation. 

5) Criteria for diagnosis of PNF should be reported in methods section and cases of delayed graft function should also be included in statistic as a separate group.

Thank you for this comment. We defined primary graft non-function (PNF) as graft failure leading to retransplantation or leading to death as judged by treating hepatologist. We clarified our definition in the “Data sources and measurements” subsection along all other outcome definitions. 

Delayed graft recovery or primary graft dysfunction is a different outcome, as mentioned by the reviewer. The most used and probably more relevant definition is the “primary graft dysfunction” definition reported by Olthoff et al (Olthoff KM, et al. Liver Transplantation. 2010;16(8):943–9). We extracted all reported outcomes from patients’ chart and adjudicated them, but we did not extract laboratory measurements within the first week after transplantation necessary to measure the incidence of such outcome in our cohort. Such outcome should probably be included in further work on the effects of liver transplantation perioperative hemodynamic management on graft outcomes. 

6) There are some inconsistencies between data in the abstract and tables or text. For instance: in abstract (results section) 562 patients are reported; in the text (data source) they seem to be 568 then in the text results section these are actually 562 liver transplants in 514 patients!? 

We included 568 liver transplantations in our study. Unfortunately, 6 patients died intraoperatively and so 6 liver transplantations could not be included in any analysis (outside survival analysis). Many patients had multiple transplantations so at the end, 519 patients were included in our study and 514 patients were included in our analyses (1 out of 6 patients who died intraoperatively had a first liver transplantation that was included in our analyses). As such, our first paragraph of the “Results” section is now: “We included 568 consecutive liver transplantations performed in 519 patients (S2 Appendix 2 - S1 Fig), of which 98% where donations after neurological determination of death (NDD). Six patients died in the operating room, so 562 liver transplantations done in 514 patients were included in the analyses (1 out of 6 patients who died intraoperatively had a first liver transplantation that was included in our analyses).” We clarified the results section of the abstract. Please let us know if any further clarification is required.

Again in the abstract 31 non anastomotic biliary strictures are reported while they seem to be 56 in Table 2 !? Please check with accuracy the reported number in abstract, text and tables. 

Thank you to have pointed out this typing error. The table presents the real results. There was in fact 56 non-anastomotic biliary strictures in our cohort. In the abstract and the result sections, the reported proportion was right (10%), but the number was the one of patients having both complications. The number has been corrected in the text and is consistent with table 2. Thank you very much to have pointed it out. 

Also in Figure S1: are these transplants or patients?

Those were transplantations. We corrected the figure. Thank you.

7) I would include figure S8 in the main text.

We moved figure S8 in the main manuscript.

---

## [Decision Letter · Decision Letter 1]

28 Jun 2021

Association between intraoperative fluid balance, vasopressors and graft complications in liver transplantation: a cohort study

PONE-D-21-17298R1

Dear Dr. Carrier,

We’re pleased to inform you that your manuscript has been judged scientifically suitable for publication and will be formally accepted for publication once it meets all outstanding technical requirements.

Kind regards,

Gianfranco D. Alpini

Academic Editor

PLOS ONE

Additional Editor Comments (optional):

Reviewers' comments:

Reviewer's Responses to Questions

**Comments to the Author**

1. If the authors have adequately addressed your comments raised in a previous round of review and you feel that this manuscript is now acceptable for publication, you may indicate that here to bypass the “Comments to the Author” section, enter your conflict of interest statement in the “Confidential to Editor” section, and submit your "Accept" recommendation.

Reviewer #1: All comments have been addressed

2. Is the manuscript technically sound, and do the data support the conclusions?

Reviewer #1: (No Response)

3. Has the statistical analysis been performed appropriately and rigorously? 

Reviewer #1: (No Response)

4. Have the authors made all data underlying the findings in their manuscript fully available?

Reviewer #1: (No Response)

5. Is the manuscript presented in an intelligible fashion and written in standard English?

Reviewer #1: (No Response)

6. Review Comments to the Author

Reviewer #1: (No Response)

7. PLOS authors have the option to publish the peer review history of their article (what does this mean?). If published, this will include your full peer review and any attached files.

Reviewer #1: No

---

## [Editor Report · Acceptance letter]

30 Jun 2021

PONE-D-21-17298R1 

Association between intraoperative fluid balance, vasopressors and graft complications in liver transplantation: a cohort study 

Dear Dr. Carrier:

I'm pleased to inform you that your manuscript has been deemed suitable for publication in PLOS ONE. Congratulations! Your manuscript is now with our production department. 

Kind regards, 

on behalf of

Dr. Gianfranco D. Alpini 

Academic Editor

PLOS ONE